RESEARCH CULTURE

# Using reflective practice to support PhD students in the biosciences

**Abstract** Postgraduate study can be mentally, physically and emotionally challenging. The levels of anxiety and depression in postgraduate students are much higher than those in the general population, and isolation can also be a problem, especially for students who are marginalised due to gender, race, sexuality, disability or being a first-generation and/or international student. These challenges are not new, but awareness of them has increased over the past decade, as have efforts by institutions to make students feel supported. Under the umbrella of a Doctoral Training Partnership, we developed a programme in which reflective practice is employed to help postgraduate students navigate work environments, deal with difficult supervisory or professional relationships, and improve their work-life balance. Additionally, this reflective practice is allowing the training partnership to tailor support to its students, enabling them to effectively nurture our next generation of bioscientists.

**JENNIFER TULLET\*, JENNIFER LEIGH, BRANDON COKE, DAVID FISHER, JOHANNA HASZCZYN, STEVEN HOUGHTON, JOHANNA FISH, LAURA FREEMAN, ISABELLA GARCIA, STEFAN PENMAN, EMMA HARGREAVES**

**\*For correspondence:** j.m.a.tullet@kent.ac.uk

**Competing interest:** The authors declare that no competing interests exist.

## Introduction

Concerns about the well-being and mental health of postgraduate students have been increasing for a number of years. One study found that postgraduate students are six times as likely to experience depression and anxiety as the general population (*Evans et al., 2018*), and another found that 30–50% of postgraduates in the United Kingdom met the thresholds for depression or anxiety (*Carr et al., 2022*). These increased concerns may be due to changes in society and academia, under-reporting in earlier studies, an increase in mental health awareness, or a combination of these factors (*Boynton, 2020*; *Dougall et al., 2021*; *Mind, 2021*).

A feeling of isolation is another problem, especially for students who are marginalised due to gender, race, sexuality, disability or being a first-generation and/or international student (*Gardner, 2008*; *Mattocks and Briscoe-Palmer, 2016*). Those who are marginalised report lower feelings of belonging, feel alone in their experiences and are often unable to reflect, process, and share what they are going through (*Harris, 2017*; *Banahene and Down, 2023*). There is a

clear relationship between mental health, loneliness and key events, such as new programmes of study (*Evans et al., 2018*). The programme we describe here addresses these situations rather than longer-term, formal diagnoses of mental health, where this approach would need to be part of a much wider set of interventions.

One approach to improving wellbeing and reducing feelings of isolation is to embed an approach called reflective practice within postgraduate programmes. Reflective practice is a way to learn from real life experiences. For example, students would be encouraged to think about their day-to-day encounters, consider how these worked, and what lessons they could take away. This article describes our experiences of running a bespoke course in reflective practice at the South Coast Biosciences (SoCoBio) Doctoral Training Partnership in the UK. This partnership includes postgraduate students at four universities – Southampton, Kent, Sussex and Portsmouth – and the National Institute for Agricultural Biotechnology.

Our approach builds on the work of the Women In Supramolecular Chemistry network

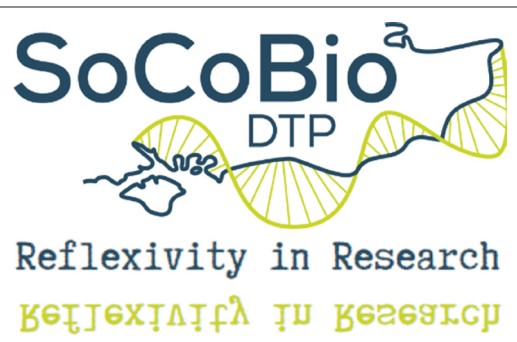

**Figure 1.** The logo for the SoCoBio Reflexivity in Research programme was co-created by students and staff. To reinforce the work the students completed during the Reflexivity in Research programme, we sent them a care package of gifts marked with the logo (such as seed packets, mugs and bags) to increase their sense of belonging to the programme, and to remind them of the benefits of reflexivity, especially when facing challenges.

(WISC; *Leigh et al., 2022*). This project involved a collaboration between researchers in the chemical and social sciences, and drew on extensive experience of working therapeutically and addressing marginalisation in science (*Egambaram et al., 2022*; *Leigh and Bailey, 2013*). In particular, it facilitated individuals to be more consciously aware of their embodied experiences (i.e., the thoughts, feelings, sensations, images and emotions they experienced in different scenarios), and to then feed these experiences into effective reflection (*Evans-Winters, 2019*; *Kujawa-Holbrook and Montagno, 2009*; *Leigh and Brown, 2021*). This included 'owning' experiences rather than projecting them onto others.

Learning to reflect is counter-intuitively hard and, in science subjects, achievement is often assumed to be directly related to coursework and exam grades rather than developing an effective practice or reflexivity (*Ixer, 1999*; *Leigh, 2016*; *Stronach et al., 2007*; *Tremmel, 1993*). When considering team dynamics and interpersonal relationships, this type of reflective practice emphasises the importance of understanding how people interact, how different people trigger different reactions, and encourages a conscious awareness of this. It is then possible to use that knowledge and insight to choose how to respond or act, rather than react, in an honest and authentic way. We built on and developed a bespoke creative reflexivity course under the umbrella of a UK government funded Doctoral Training Partnership, aimed specifically at bioscientists.

## The South Coast Biology Doctoral Training Partnership

The South Coast Biosciences Doctoral Training Partnership (SoCoBio DTP) has been funded by the Biotechnology and Biological Sciences Research Council (BBSRC). SoCoBio students hold a PhD studentship at one of five different UK institutions (the Universities of Southampton, Kent, Sussex and Portsmouth, as well as National Institute for Agricultural Biotechnology (NIAB) at East Malling). The first cohort of SoCoBio DTP students were recruited to their programme in early 2020 meaning that the COVID-19 pandemic lockdowns and restrictions in the UK prevented in-person induction events, and greatly restricted lab work at the start of their PhDs. This cohort, and the next three cohorts, have experienced a variety of challenging and rapidly changing working conditions in addition to the usual demands of postgraduate training.

Given this challenging and unusual start combined with the national statistics on postgraduate wellbeing, the SoCoBio management board were keen to learn more about the wellbeing of students on the programme so they could support them appropriately. All students are asked about their general wellbeing, mental health, the types of challenges they face, and how these impact their work in a yearly survey (*Supplementary file 1*). Over half the survey participants (54% (2021), 62% (2022) and 65% (2023)) reported experiencing mental health or general wellbeing difficulties each year. In 2021, it was notable that over one in three of the students reported difficulties attributed to the COVID-19 pandemic, whilst by 2022 this was reduced to less than one in five. Of those experiencing challenges, around 35% reported this had affected their ability to carry out their work. These figures, although concerning, suggest that the wellbeing of students enrolled in the SoCoBio DTP is similar to that of UK postgraduate students in general (*ONS, 2021*).

In the annual survey, our students reported that the challenges impacting their wellbeing were wide-ranging. Whilst some were of a personal nature, challenges faced in the context of their PhD environment could be broadly categorised into: technical problems (experimental design, reproducibility, learning new techniques etc.); concerns about achievement or generating results; the jump from undergraduate to postgraduate study; negotiating and managing supervisor relationships; work-life balance; and a general feeling of not belonging or fitting

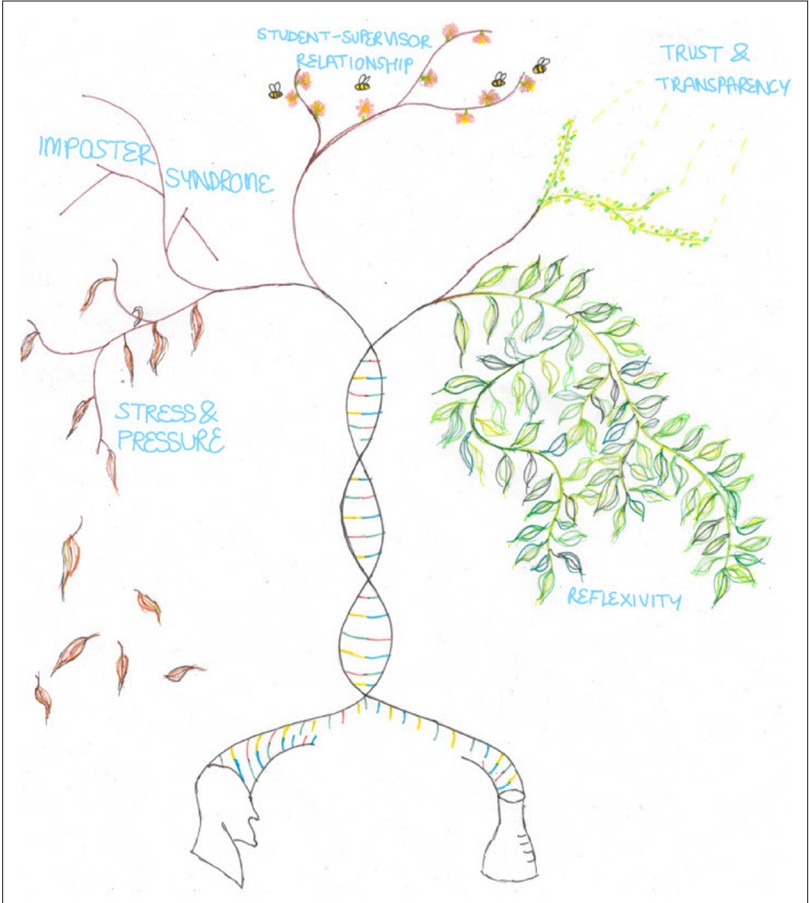

**Figure 2.** Important themes for postgraduate students. The themes that emerged following a thematic analysis of transcripts from the pilot sessions of the Reflexivity in Research programme were: stress & pressure; imposter syndrome; student-supervisor relationship; trust & transparency; and reflexivity. Image drawn by one of the participants. The participants were very clear that the programme was highly beneficial to navigating the challenges of postgraduate study.

into research culture. They were encouraged to discuss any challenges with their supervisor, but we also recognised the potential benefit (to both students and supervisors) of an additional forum for students to build a sense of community, discuss wellbeing, and learn to deal constructively with common challenges. We wanted to find an approach that would support and nurture the development of students, give them space and time to address their concerns, and support diversity and inclusivity within the programme.

## The SoCoBio Reflexivity in Research programme

Addressing wellbeing and belonging for postgraduate students is a powerful tool to mitigate against increased attrition rates, and increase diversity in science: this is important for individual students, their supervisors, Universities, and (in the context of government funded research) the taxpayer. Given the national data on postgraduate wellbeing and the compounding fact that our doctoral training Partnership was initiated in parallel with an international pandemic, we wanted to ensure that we took care of the next generation of UK trained bioscientists. Reflexivity in Research was developed as a six-month online programme (six sessions) for students in the second year of their PhD.

The pilot and evaluation were given ethical approval to use anonymised data from participating students by the University of Kent. Eight participants (four women and four men, all UK home students, representing ~a third of the 2020 cohort) volunteered to participate and were randomly divided into two groups. The students co-created a logo, providing the programme with an identity (*Figure 1*); and all participants were sent a care package of gifts marked with the programme logo that they could use in their daily lives. These gifts acted as a reminder of their participation and fostered feelings of connection and belonging (*UKRI, 2023*).

The Reflexivity in Research programme encouraged individuals to reflect on their professional identity and inter-personal relationships under the guidance of a trained academic facilitator (*Kara, 2022*). Creative and reflective approaches were designed to facilitate emotional engagement and allow participants to access authentic stories, and as such needed to be facilitated with care: thus, the facilitators were integral to the programme. They organised meetings, set tasks prior to each session, and facilitated discussion. The facilitators were all experienced academics who had personal experience of a scientific doctoral programme and an interest in promoting postgraduate wellbeing.

Initially all facilitators were from the University of Kent. Facilitators were not directly involved in the doctoral training of their participants and although PhD supervisors can be trained as facilitators, they should not lead a group with students that they supervise. Each facilitator attended a programme of sessions on reflexivity and creative research techniques with a reflexivity expert and trained therapist. This ensured that they were aware of how to hold the boundaries of the programme, and their role to support students and signpost them to more specialist therapeutic services where appropriate. The latter point is important as although none of the facilitators were trained counsellors or therapists,

> ## Box 1. Topics that emerged in discussions of reflexivity in the biosciences, and the opportunities they present.
>
> - **Who are you as a bioscientist?** What does it mean to be a bioscientist? What do you love about your subject? How do you fit into your workplace? What are your concerns? An opportunity for the group to get to know each other and discuss belonging and imposter syndrome.
> - **Expectations of life as a PhD student.** What do you expect from yourself and what do you think others (supervisor, family, friends etc) expect of you? A space to share any concerns about professional or personal relationships that may impact PhD progress and gain peer support. This also sparked discussions about the importance of tenacity.
> - **Pressure and stress management.** How do you feel when you are under pressure or stress and how do you manage these feelings? Discussing this offers opportunities to see how others are affected by and manage stressful situations, discuss work-life balance, and for advising on coping mechanisms and signposting to appropriate service within institutions.

work of this kind can easily border onto therapeutic processes.

As part of this training, facilitators explored the same questions and prompts so that they could explore their own emotional responses and stories and choose what they wished to share. The importance of peer support was also built into the programme and facilitators regularly met with a dedicated expert supporter of the programme to share their experiences and provide support for each other. This was particularly important for protecting the academics' mental health as well as providing useful ways to optimise the programme.

The content of the meetings was not focused on academic attainment and progress, and as such was separate from research group meetings and academic adviser or tutorial meetings (*Kara, 2015*). It was important that the reflective sessions occurred in as safe an environment as possible to encourage emotional participation and openness. Students were encouraged to find a quiet, private space and were assured that all discussions were confidential and that any notes taken during the sessions would be securely stored and protected. Each meeting had a prompt or question to address, with students asked to reflect in advance and bring something which they would like to share and discuss. Topics included: What qualities are needed to be successful and when did you notice these in yourself? and How do you respond to stress or pressure? In all cases students were encouraged to write, mark-make, draw, or find an image or object to represent how these situations made them feel (*Box 1*). The

programme was evaluated using pre- and post-online surveys, analysis of transcribed recordings of the meetings, and reflective notes made by the facilitators. The students and facilitators also co-created the end evaluation questions. The key factors for implementing a successful programme of Reflexivity in Research in the biosciences are outlined in *Box 2*.

The information gained from these two-way interactions between academic facilitators and PhD students throughout the programme is now allowing the DTP management board to develop support and training structures for both students and supervisors. Indeed, the students undertaking the programme believed that teaching supervisors about the reflexivity programme would be beneficial. This is being undertaken as part of a wider drive in the UK to improve supervisor training (*UKRI, 2023*; *Wellcome Trust, 2023*). Additionally, inviting students to participate in this programme as part of their core PhD training ensures that the skills required for personal reflection and adaptation are embedded from the initial stages of their postgraduate degree. Therefore, it can provide them with a lifelong toolkit specifically designed to support scientific work. That said, we recognise that some of the problems students are facing in academia cannot be fixed using this approach and require greater institutional and societal commitment. Marginalised students face systemic barriers which can include a lack of representation in their peers or supervisors and lack of access to opportunities and networks (*Mattocks and Briscoe-Palmer, 2016*; *UKRI, 2023*; *Cech, 2022*). Dismantling

> ## Box 2. Examples of good practice when implementing reflexivity, and advice for facilitators.
>
> **Examples of good practice**
> **Training:** Sessions must be carried out by an academic trained to be a reflexivity facilitator.
> **Impartiality:** Reflexivity facilitators must be impartial.
> **Trust:** Sessions must foster a sense of trust and belonging to encourage emotional participation.
> **Confidentiality:** Sessions must be confidential.
> **Peer support:** The reflexivity facilitators need to support each other within or across institutions and meet regularly to share experiences and ideas.
>
> **A facilitator should:**
>
> - Present their ideas first to demonstrate different types of creative and reflective approaches, and to build trust.
> - Be ready for any topic of discussion.
> - Set expectations for attendance and engagement early on and encourage full participation.
> - Make sure the whole group is heard and thank participants for sharing their experiences.
> - Identify and implement ways of assessing students' expectations upon entering the programme and their feelings on completing it.
> - Ensure that there is time to reflect on the sessions and to receive support and advice from other facilitators, particularly if challenging discussions or circumstances arise during the sessions (see *Box 3*).

these barriers will require structural and societal change, so while this programme can support students with the situation they are in, it cannot be used as a sticking plaster to support a flawed model.

Following the six reflexivity sessions, the recordings of the meetings were transcribed, and a reflexive thematic analysis was undertaken (*Braun and Clarke, 2021*). The analysis was undertaken by hand (*James, 2013*), and themes or areas of interest were identified by each facilitator (*MacLure, 2003*). These were then discussed by the research team and the student participants to define the five themes discussed below. (*Figure 2*).

## The five themes to emerge from the SoCoBio Reflexivity in Research programme

### Theme 1: Coping with stress and pressure

The programme was explicitly designed to support the participants to reflect on how they coped with stress and to explore the mechanisms they could put into place to support themselves and each other. This recognised that postgraduate study is often a time of intense pressure and stress (*Ayres, 2022*). Establishing an academic identity within the current academic environment in any discipline demands that an individual deals with the pressure and stress (*Clark and Sousa, 2018*; *Gill, 2009*). However, those who are marginalised in science are much more likely to face stress (*Rolle et al., 2021*). All the student participants recognised that the scientific research environment brings unique challenges. They had anticipated some of these challenges but, others were completely unexpected. When asked to draw or find an image to represent what stress felt like to them; one analogy used was the comparison of stress to a tidal wave, i.e., you either succumb to that wave and go under, or drag yourself out and persevere. The unexpected challenges particularly caused the students to report feeling overwhelmed, stressed, anxious, and frustrated. Some situations that caused stress and anxiety related to daily challenges of experimental work, but others were centred on personal relationships. When discussing the former, one

## Box 3. How to set up a reflexivity programme.

- Engage your institution in the programme, encourage them to define and reward your time commitment.
- Identify facilitators and train them in reflective practice.
  - Facilitators need to engage in the creative and descriptive aspects of reflective practice, and to demonstrate this to their group. Reflexivity can border on therapeutic practice and it is critical that facilitators are prepared.
- It can be helpful if the facilitator and the students work in similar scientific disciplines, particularly as different postgraduate programmes can offer very different challenges.
  - Facilitators must not be directly related to the student's PhD project.
- Map a support network for the facilitators. This could be regular meetings with other facilitators, as well as clear links to the relevant head(s) of graduate studies, and to the staff responsible for wellbeing and student support at your institution.
- Identify the cohort you want to help; clarify who will have access to the programme; determine how long the programme will run, given the resources available. Small group sessions of ~6 students work best.
- Be organised! Engage your cohort from the start of your programme, explain it is a pilot, and involve them in your planning and evaluation processes.

participant described the experimental challenges they were facing and their feelings when talking about them with their supervisor.

> 'If you get things working first time it is great. Everyone's like, great, amazing. You know, this is something we can now do. If it doesn't work, you almost imagine being sat in a police interview.' **Graduate student, man.**

To balance out the discussions of stress, participants were asked to share the mechanisms they used to support themselves. These were varied but included making time for exercise, reaching out to family and friends, and treating themselves to good food. The students were all aware of the mechanisms that they could use to support themselves but also vocalised the difficulties with applying them to themselves and balancing them with their work. The programme was focused on facilitating effective reflective practice and reflexivity rather than rumination, which is a maladaptive coping strategy linked to depression and anxiety (*Joireman et al., 2002*). Interestingly, other maladaptive coping strategies such as alcohol or drug use were not discussed by the participants, but it is worth noting that these may be raised in future cohorts and facilitators should be prepared to provide clear and supportive signposting.

> 'I tried so many different things to try and ease stress, and I just haven't found anything that I feel works. I've tried taking a few days out and going to have fun and doing stuff, but, then I just feel guilty for not working. I know things which could possibly help, exercise, spending time with family, partners, friends going outside, activities. … I prioritise work very much over those things. And, if I do make time for them, then I find myself feeling guilty about not doing work.' **Graduate student, woman.**

The groups were encouraged to listen to each other and recognise how individuals deal and respond to stress differently, so that they might formulate a toolkit to address their own levels of stress to prevent burnout. This stimulated discussions on the importance of work life balance, informal support networks including peers, supervisors, colleagues, friends and family, and offered a platform to outline more formal support available for postgraduate students (such as Departmental and University wellbeing and mental health services, and local charities).

### Theme 2: Challenges of the student-supervisor relationship

It has long been recognised that supervisory relationships can be challenging (*Eshtiaghi et al., 2012*). One key aspect is the 'fit' between student and supervisor and the quality of the relationship between them (*Gill and Burnard, 2008*; *Johansson and Yerrabati, 2017*; *Löfström and Pyhältö, 2017*; *Sambrook et al., 2008*). Several students in the groups were experiencing issues with their supervisor. They described themselves using words such as 'lost', 'alone', 'overwhelmed', 'tired', 'unmotivated', 'frustrated' and 'annoyed'. It is easy to see how a student caught up in this storm of negative emotion might not feel they belong in science (*Royal Society of Chemistry, 2021*). The programme provided a space and facilitated students to reflect on and articulate the challenges they were facing in their supervisory relationships. The facilitators were able to navigate discussions on the expectations of a supervisor, a PhD, and allowed the groups time to reflect on this:

> 'I would like to have a clearer understanding what is expected of me as a PhD student. I feel that I put a lot of pressure on myself and take on too much, meaning I work almost constantly and have very little free time to spend enjoying myself and being with others.' **Graduate student, woman.**

> 'What was expected of me wasn't clear: In a normal 9–5 working world the boss says, do X, Y, Z. You do X, Y, Z and know you're on track. Whereas with a Ph.D. you don't always get that kind of feedback. It surprised me how self-led it was. Am I on track or falling behind? Or am I doing enough? Am I doing too much or overworking myself? Or am I doing OK? It is hard to gauge and I am having to figure that out myself.' **Graduate student, man.**

> '[What I am finding is] that the nature of a PhD it is very much self-led, it's self-managed and one of the core things I've learned is you can get out what you put in.' **Graduate student, woman.**

The students highlighted the importance of clear communication, regular feedback, and transparency around the roles and expectations of both a student and supervisor through a PhD programme.

### Theme 3: Dealing with imposter syndrome and maintaining authenticity

Imposter syndrome is rife within academia (*Bothello and Roulet, 2019*; *Taylor and Lahad, 2018*) and not addressing it threatens the aim to increase diversity and inclusion in science (*Chrousos and Mentis, 2020*). Unsurprisingly, many of the participants expressed general feelings of not fitting in within the research environment. One of the first exercises they were asked to complete was to reflect on how they saw themselves as a scientist, and what being a scientist meant.

> 'It's really hard because I don't think I'm a scientist. I know I'm in the lab doing work and I'm pipetting and, you know, doing all that stuff. But I don't feel like a scientist even if I am in a white lab coat.' **Graduate student, man.**

At the same time, students shared the desire to remain true to their own identify and values and did not wish to conform to a scientific stereotype. This also led to discussions about the importance of diversity within research groups and valuing and understanding an individual's contributions.

### Theme 4: Building trust and transparency in professional networks

Trust was a strong theme throughout and came up in almost all aspects of our discussions. This included trust in peers, laboratory colleagues, supervisors, the DTP itself and their personal relationships. All of these were important for maintaining a sense of balance and security in a rapidly changing and challenging research environment. The concept of relating a beautiful, well-tended garden to the PhD environment was used by several students:

> 'PhD students are the flowers and plants here and have an expectation that they will grow and flourish to the best of their ability. They would try and support one another and be within this community, amongst other like-minded people and other PhD students. But of course, they need nourishment, and the expectation is, of course, that for this beautiful garden to exist, you need the support of gardeners (our supervisors and the wider DTP cohort).' **Graduate student, man.**

The students were very engaged with the programme (attendance rate of >85%) and willing to discuss how they can support themselves and their peers both through engaging

in the Reflexivity in Research programme, and via the DTP. It was clear from the outset that students feel that transparency and inclusion in DTP processes were also very important. Making it clear that students are important, included and valued at all levels in the DTP will help to foster a more inclusive and open environment.

### Theme 5: The benefits of the Reflexivity in Research programme

We received overwhelmingly positive feedback from the students on the programme.

> 'It has helped me mentally and to gain perspective. Imposter syndrome is a big part of my struggle with PhD life, but reflexivity has helped me try and think differently about these thoughts. I am trying to put the session discussions into practice and hopefully become slightly more confident in myself and help me in my PhD mind-set.' **Graduate student, man.**

> 'It has made me more resilient. I think I am more likely to be kind to myself, and happy with my work, because I will have done my best, and my best is good enough.' **Graduate student, woman.**

> 'I feel I have learned to look at my emotions differently when confronted with lab failure or overwhelming stress. This has made me a more efficient and compassionate scientist; I hope to encourage others to do the same.' **Graduate student, woman.**

Students from the pilot also formed their own networks on social media and have remained in touch, demonstrating the power of emotional connection when forming networks and the sense of belonging created by the programme.

As a prompt in the last reflexivity session, the participants co-created a set of evaluation questions that could be used for this and future cohorts. Following the thematic analysis of transcripts, the participants also verified the themes arising and were involved in the production of this manuscript.

Finally, we sought feedback from colleagues impacted by this work – management board members of the DTP, PhD supervisors of participating SoCoBio students and facilitators – to determine their perceptions of the programme and its usefulness.

> 'Our SoCoBio student cohorts have faced unprecedented challenges during either

their undergraduate and postgraduate studies caused by the Covid-19 pandemic on top of the normal issues associated with PhD study. This has resulted in an impact [on] their mental health and it is fantastic that we have this programme to offer them by way of support. By tailoring it to the shared experiences of SoCoBio students it has been able to provide help that is directly relevant to the lives of our student cohorts and those that have taken advantage of this opportunity so far have clearly benefitted. SoCoBio will continue to support this endeavour to make it more widely available to our own students and those on other programmes.' **Professor Matthew Terry, Director of the SoCoBio DTP**

> 'Being a trained 'Reflexivity in Research' facilitator has had an immeasurable impact on my understanding of how we can support PhD students to become effective researchers outside of traditional supervisory relationships. Over the 6 sessions it was incredible to see the group evolve and engage in open and transparent conversations.' **Dr Emma Hargreaves, trained Reflexivity facilitator**

> 'This is a fantastic opportunity to learn and develop with each other, building the sense of comradery that is easy to forget as we work on individual projects. While as a supervisor, I am always willing and available to help, I understand and appreciate the importance of peers to help cope with and process the challenges of postgraduate life.' **Anonymous, Supervisor of SoCoBio student.**

This positive feedback supports and validates the success of the Reflexivity in Research programme. It is clear that these academics both recognise a need for this type of programme and can see its benefits. This positive feedback strengthens our incentive to continue developing the programme in collaboration with both students and academics.

## Discussion

Moving forwards, we will embed reflexivity into the SoCoBio Doctoral Training Programme. New facilitators have been trained, allowing for increased capacity for students and balance of

academic workloads. Currently, these new facilitators are all from the University of Kent, but from 2024 this will be expanded to other institutions in the partnership. So far, all facilitators have been white female academics, but we are actively encouraging colleagues from all genders and backgrounds to volunteer. Historically, female academics have tended to take on more pastoral roles (*Rosser, 2004*) but a diversified approach will improve equality within the programme, providing role models who will share their diverse experiences to inspire our next generation of bioscientists.

> *'As an academic it can be daunting to tread the boundary between academic supervision and supportive friend during challenging circumstances. Reflexivity training has helped me understand and build my confidence in the role(s) academics can play to support students (in addition to guiding them to formal mental-health support) whilst still defining boundaries and forming strong mentoring relationships with the students.'* **Dr Jenny Tullet, PhD supervisor and trained Reflexivity facilitator.**

The evaluation provided us with a deeper and more accurate understanding of the challenges faced by our students, and provides a basis of knowledge with which to devise strategies to build into the DTP programme to support them in future. The programme described will now run every year with a cohort of second-year students, supplemented with top-up sessions at other points during the DTP training schedule (e.g., during the induction event for new starters, and the annual conference). These sessions will advertise and provide understanding of the Reflexivity in Research programme throughout PhD training. We will continue to use student feedback, and will add biannual top-up sessions (for graduates of the programme) to encourage reflexivity as a life-long practice, and will allow students reflective time throughout their PhD. Additionally, this feedback will be used to direct DTP administrative processes and other wellbeing initiatives to create tools and processes that engage and direct students facing challenges. For instance, participating students suggested incorporating elements of reflexivity into our supervisor training sessions as a way of allowing supervisors to understand the benefits of this approach and encourage them to support students in dedicating time to these activities. Together, we hope that these strategies will promote feelings of

acceptance, belonging, and positive wellbeing within our postgraduate cohorts.

We are also working to share good practice beyond our own DTP and have taken reflexivity-based events to conferences, including the British Society of Research into Ageing annual scientific meeting in 2022. The independent success of our programme and the reflexivity work of the WISC network in chemistry (*Caltagirone et al., 2021*; *Leigh et al., 2022*) suggests that this activity would translate well to other disciplines within the natural sciences. It would fit well with funders' commitments to increase diversity and improve the mental health and wellbeing of postgraduate researchers (*UKRI, 2023*; *Wellcome Trust, 2023*) as it has been designed to address issues – such as feeling isolated and a lack of belonging **–** that impact marginalised students to a greater extent. Our vision is to embed reflexivity in postgraduate research programmes, both nationally and internationally, and *Box 3* contains advice on how to set up a reflexivity programme in the biosciences.

## Acknowledgements

We thank the students who participated in this pilot for their willingness to openly share their experiences; Professor Jennifer Hiscock for the inspiration to start this project; Professors Daniel Osario, Sarah Guthrie and Matthew Terry for support and encouragement in the initial stages; and Professor Matthew Terry, Dr Jill Shepherd and Dr Marina Ezcurra for critically reading the manuscript. This work was supported by BBSRC EDI Implementation Funding included in BB/T008768/1.

**Jennifer Tullet** is in the Division of Natural Sciences, University of Kent, Canterbury, United Kingdom
j.m.a.tullet@kent.ac.uk
https://orcid.org/0000-0002-2037-526X

**Jennifer Leigh** is in the School of Sociology, Social Policy, and Social Research, Division for the study of Law, Society, and Social Justice, University of Kent, Canterbury, United Kingdom
https://orcid.org/0000-0002-3672-1462

**Brandon Coke** is in the School of Biological Sciences, University of Southampton, Southampton, United Kingdom
https://orcid.org/0000-0002-0847-6885

**David Fisher** is in Crop Science and Production Systems, National Institute of Agricultural Botany, East Malling, United Kingdom
https://orcid.org/0000-0002-7947-3366

**Johanna Haszczyn** is in the School of Biological Sciences, University of Southampton, Southampton, United Kingdom
https://orcid.org/0000-0002-7012-7125

**Steven Houghton** is in the School of Biological Sciences, University of Southampton, Southampton, United Kingdom

🔟 https://orcid.org/0000-0002-9568-3769

**Johanna Fish** is in the School of Biological Sciences, University of Southampton, Southampton, United Kingdom

🔟 https://orcid.org/0000-0001-5683-9946

**Laura Freeman** is in the Division of Natural Sciences, University of Kent, Canterbury, United Kingdom

🔟 https://orcid.org/0000-0003-1674-8112

**Isabella Garcia** is in the Division of Natural Sciences, University of Kent, Canterbury, United Kingdom

🔟 https://orcid.org/0009-0008-8208-8355

**Stefan Penman** is in the School of Life Sciences, University of Sussex, Brighton, United Kingdom

**Emma Hargreaves** is in the Division of Natural Sciences, University of Kent, Canterbury, United Kingdom

🔟 https://orcid.org/0000-0001-7887-2018

*Author contributions:* Jennifer Tullet, Conceptualization, Data curation, Formal analysis, Supervision, Funding acquisition, Investigation, Methodology, Writing – original draft, Project administration, Writing – review and editing; Jennifer Leigh, Conceptualization, Formal analysis, Supervision, Methodology, Writing – review and editing; Brandon Coke, Investigation, Writing – review and editing; David Fisher, Investigation, Writing – review and editing; Johanna Haszczyn, Investigation, Writing – review and editing; Steven Houghton, Investigation, Writing – review and editing; Johanna Fish, Investigation, Writing – review and editing; Laura Freeman, Investigation, Writing – review and editing; Isabella Garcia, Investigation, Writing – review and editing; Stefan Penman, Investigation, Writing – review and editing; Emma Hargreaves, Conceptualization, Data curation, Formal analysis, Supervision, Investigation, Methodology, Project administration, Writing – review and editing

*Competing interests:* The authors declare that no competing interests exist.

## Funding

| Funder | Grant reference number | Author |
| --- | --- | --- |
| Biotechnology and Biological Sciences Research Council | EDI Implementation Funding BB/T008768/1 | Jennifer Tullet |

The funders had no role in study design, data collection and interpretation, or the decision to submit the work for publication.

## Decision letter and Author response

Decision letter https://doi.org/10.7554/eLife.92365.sa1
Author response https://doi.org/10.7554/eLife.92365.sa2

## Additional files

### Supplementary files

• Supplementary file 1. The SoCoBio Wellbeing Survey April 2023.

### Data availability

No new data were generated for this article.

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
