## [Decision Letter]

**Decision letter after peer review:**

Thank you for submitting your article "Using reflective practice with PhD students to build personal and professional resilience" to *eLife* for consideration as a Feature Article. Your article has been reviewed by three peer reviewers, and the evaluation has been overseen by two members of the *eLife* Features Team (Hazel Walker and Peter Rodgers). The following individuals involved in review of your submission have agreed to reveal their identity: Louise Banahene and Joanna Royle; the third reviewer prefers to remain anonymous.

The reviewers and editors have discussed the reviews and we have drafted this decision letter to help you prepare a revised submission.

Summary

This useful paper presents a valuable case study on the creation of a bespoke reflective practice initiative to support the personal development of postgraduate researchers at a cross-institutional Doctoral Training Partnership. It offers helpful models for practice to other institutions and research groups seeking to embed reflective practice, with highly transferable advice that could be used with other cohorts and programme designs. Strengths include grounding 'Reflexivity in Research' in an evidence base about the needs of the cohort; getting researchers involved in co-design from the start; care taken around who connects to who (separating out facilitators from candidates whose doctoral journey they are involved in); facilitator training and peer support, and evaluation built in at the point of design. However, there are a number of concerns that need to be addressed, as outlined below.

Essential revisions

1. Assuming that the data are available, it would be helpful to include information on the diversity of the cohort available (e.g. gender, UK vs non-UK, number of students from marginalized groups). This is relevant given that the rationale references the extent to which marginalized students are more likely to be affected by belonging or mental health.

2. The manuscript rightly references the impact of belonging and mental health on experiences of postgraduate study. This could be strengthened by outlining the inter-relationship between mental health, loneliness and key events, such as new programmes of study. This would help to clarify the relevance of the intervention. Without this there is a danger that the reader may conflate the term mental health, with longer-term, formal diagnoses, where this approach would need to be a part of a much wider set of interventions.

3. In addition, whilst there is acknowledgment that the intervention cannot address all the barriers a student may experience there would be benefit in referencing the structural barriers that may exist for some students (including those from marginalised communities). There is a need to address these through structural change alongside interventions for students. This addition will also help to ensure that readers are clear that this is not an intervention that is reinforcing a deficit model.

4. Please include a Methods section at the end of the article that covers data collection, data analysis and interpretation, and ethical considerations (eg, did the research require institutional approval?). Please also include a copy of the annual survey as an additional file.

Also, when the text states that students reported wide ranging challenges, it is not clear if this information was generated through interview/focus group/open-ended survey questions: please clarify

5. Please be cautious about the implications of using the word 'resilient' in the article title. In academic literacies and researcher development disciplines resilience is not infrequently associated with toxic positivity and this may lead to the paper being unfairly harshly judged by readers.

6. Reference is made to the programme "allowing the DTP…to target… supervisor training". There is nothing else in the paper that mentions supervisor training. Supervisor training is a major element of, for example, the recently published UKRI New Deal, so if it is not going to be explored in detail, it might be best to drop it.

7. Various parts of the article would benefit from further discussion (and citation) of previous work, as outlined below:

7.1. All contextual and methodological claims in the article should be supported by relevant references. Two examples are the 'Good Practice for Reflexivity', and the claim that "female academics have tended to take on more pastoral roles"

7.2. When the authors write that they "drew on extensive experience of working therapeutically and addressing marginalisation in science" and "facilitated individuals to be more consciously aware of their embodied experiences", they discuss/cite only their own work: please consider discussing/citing the work of other groups if possible.

7.3. The paper does not sufficiently engage with the existing literature on reflective practice and the ethics of care in HE.

7.4. While there are some methods references, the paper would be improved by further engagement with the literature on co-creative research methods, ethnography, qualitative thematic analysis etc.

---

## [Author Response]

SummaryThis useful paper presents a valuable case study on the creation of a bespoke reflective practice initiative to support the personal development of postgraduate researchers at a cross-institutional Doctoral Training Partnership. It offers helpful models for practice to other institutions and research groups seeking to embed reflective practice, with highly transferable advice that could be used with other cohorts and programme designs. Strengths include grounding 'Reflexivity in Research' in an evidence base about the needs of the cohort; getting researchers involved in co-design from the start; care taken around who connects to who (separating out facilitators from candidates whose doctoral journey they are involved in); facilitator training and peer support, and evaluation built in at the point of design. However, there are a number of concerns that need to be addressed, as outlined below.

Thank you for the positive response to our manuscript. We value your thoughtful feedback and have endeavored to address your concerns below.

Essential revisions1. Assuming that the data are available, it would be helpful to include information on the diversity of the cohort available (e.g. gender, UK vs non-UK, number of students from marginalized groups). This is relevant given that the rationale references the extent to which marginalized students are more likely to be affected by belonging or mental health.

We have now added further information to the manuscript on gender balance and nationality. We did not ask the students about marginalization, and do not have ethical approval to divulge any further information in this format.

“8 participants (4 women and 4 men, all UK home students, representing ~a third of the 2020 cohort) volunteered to participate and were randomly divided into two groups.”

2. The manuscript rightly references the impact of belonging and mental health on experiences of postgraduate study. This could be strengthened by outlining the inter-relationship between mental health, loneliness and key events, such as new programmes of study. This would help to clarify the relevance of the intervention. Without this there is a danger that the reader may conflate the term mental health, with longer-term, formal diagnoses, where this approach would need to be a part of a much wider set of interventions.

This is an important point. We have added the following statement, including an additional reference, to paragraph 2:

“There is a clear relationship between mental health, loneliness and key events, such as new programmes of study (Evans et al., 2018). The programme we describe here addresses these situations rather than longer-term, formal diagnoses of mental health, where this approach would need to be part of a much wider set of interventions.”

3. In addition, whilst there is acknowledgment that the intervention cannot address all the barriers a student may experience there would be benefit in referencing the structural barriers that may exist for some students (including those from marginalised communities). There is a need to address these through structural change alongside interventions for students. This addition will also help to ensure that readers are clear that this is not an intervention that is reinforcing a deficit model.

We have added the following text and references:

“Marginalised students face systemic barriers such including a lack of representation in their peers or supervisors and lack of access to opportunities and networks (Mattocks et al., 2016 and Post 2023 and Cech 2022). These will require structural and societal change so, while this programme can support students within the situation they are in, it cannot be used as a sticking plaster to support a flawed model.

4. Please include a Methods section at the end of the article that covers data collection, data analysis and interpretation, and ethical considerations (eg, did the research require institutional approval?). Please also include a copy of the annual survey as an additional file.Also, when the text states that students reported wide ranging challenges, it is not clear if this information was generated through interview/focus group/open-ended survey questions: please clarify

We agree that this information would be helpful to the reader. In the paper we stated the ethical approval provided:

“The pilot and evaluation were given ethical approval to use anonymized data from participating students by The University of Kent.

We also discussed the types of questions asked in the yearly survey:

“All students are asked about their general wellbeing, mental health, the types of challenges they face, and how these impact their work in a yearly survey. Over half the survey participants (54% (2021), 62% (2022) and 65% (2023)) reported experiencing mental health or general wellbeing difficulties each year. In 2021 it was notable that over one in three of the students reporting difficulties attributed to the COVID19 pandemic, whilst by 2022 this was reduced to less than one in five. Of those experiencing challenges, around 35% reported this had affected their ability to carry out their work. These figures, although concerning, suggest that the wellbeing of students enrolled in the SoCoBio DTP is similar to that of UK post-graduate students in general (ONS.gov.uk, 2021)”.

The reviewer has correctly surmised that the challenges that we refer to were identified from our annual survey. However, we realise that we could be clearer on this point and have amended the wording as follows:

“In the annual survey our students reported that the challenges impacting their wellbeing were wide ranging.”

we are also now providing a copy of the annual survey for the years reported in Supplementary file 1 and refer to this in the text.

“All students are asked about their general wellbeing, mental health, the types of challenges they face, and how these impact their work in a yearly survey (Supplementary file 1)”.

5. Please be cautious about the implications of using the word 'resilient' in the article title. In academic literacies and researcher development disciplines resilience is not infrequently associated with toxic positivity and this may lead to the paper being unfairly harshly judged by readers.

Thank you for raising this point, in the original version of the manuscript we had meant the phrase in a positive sense but see that it can easily be misconstrued. To address this we have changed the title to: “Using reflective practice to support Bioscience PhD students”; and removed all use of the word “resilience” in the manuscript. The only exception to this was in a quote from a student who described the programme as “making them more resilient”, and we did not wish to change their words.

6. Reference is made to the programme "allowing the DTP…to target… supervisor training". There is nothing else in the paper that mentions supervisor training. Supervisor training is a major element of, for example, the recently published UKRI New Deal, so if it is not going to be explored in detail, it might be best to drop it.

The Reflexivity in Research programme was designed to encourage individuals to reflect and think about who they are as a scientist and become more aware of their professional and inter-personal relationships. This is helping the DTP management to better understand it’s cohort’s needs. In fact, at the end of the programme the students told us that they thought integrating reflexivity and knowledge of the programme into staff training would be beneficial to them. To clarify further we have re-worked the section on this to read:

“The information gained from these two-way interactions between academic facilitators and PhD students throughout the programme is now allowing the DTP management board to develop support and training structures for both students and supervisors. Indeed the students undertaking the programme believed that teaching supervisors about the reflexivity programme would be beneficial. This is being undertaken as part of a wider drive in the UK to improve supervisor training (UKRI, 2023; WellcomeTrust, 2023).”

7. Various parts of the article would benefit from further discussion (and citation) of previous work, as outlined below:7.1. All contextual and methodological claims in the article should be supported by relevant references. Two examples are the 'Good Practice for Reflexivity', and the claim that "female academics have tended to take on more pastoral roles"7.2. When the authors write that they "drew on extensive experience of working therapeutically and addressing marginalisation in science" and "facilitated individuals to be more consciously aware of their embodied experiences", they discuss/cite only their own work: please consider discussing/citing the work of other groups if possible.7.3. The paper does not sufficiently engage with the existing literature on reflective practice and the ethics of care in HE.7.4. While there are some methods references, the paper would be improved by further engagement with the literature on co-creative research methods, ethnography, qualitative thematic analysis etc.

We apologise for neglecting this in the original manuscript. We were drawing on our own professional and personal experiences of reflexivity as well as trying to limit the word count. The literature on reflexivity, ethnography, qualitative thematic analysis, and co-creative research methods in STEM disciplines is limited. However, where possible we have now added the following references to support our statements on some of these topics in order to place it in a wider context:

Good practice for reflexivity: We felt that this was well covered with the 3 references used (Ixer, 1999; Leigh, 2016 and Tremmel, 1993) that cover our work in Chemistry as well as how reflexivity is used effectively in social work and teacher education. But have now added Stronach et al., 2007 as a more general introduction to the subject. To further address the ethics of care in Higher Education including methods of working and co-creative approaches we have added the following.Pastoral roles of female academics: We had added in reference to Rosser 2004.Marginalisation in science: In this revision, in response to point 3, we added additional references Mattocks et al., 2016 and Post 2023 and Cech 2022 and some further discussion on this topic.

“Marginalised students face systemic barriers such including a lack of representation in their peers or supervisors and lack of access to opportunities and networks (Mattocks et al., 2016 and Post 2023 and Cech 2022). These will require structural and societal change so, while this programme can support students within the situation they are in, it cannot be used as a sticking plaster to support a flawed model.

Additional references referring to embodied experiences have been added:

“In particular, it facilitated individuals to be more consciously aware of their embodied experiences (i.e., from the thoughts, feelings, sensations, images and emotions they experienced in different scenarios), and to then feed these experiences into effective reflection (Evans-Winters, 2019; Kujawa-Holbrook and Montagno, 2009; Leigh J, 2021).”